# FULLY FINE-TUNING BEATS PARAMETER EFFICIENT FINE-TUNING FOR CLIP IN DATA-LIMITED SCENARIOS

## ABSTRACT

Prompt tuning, which involves training a small set of parameters, effectively enhances the pre-trained Vision-Language Models (VLMs) to downstream tasks. However, this approach often comes at the cost of flexibility and adaptability when the tuned models are applied to different datasets or domains. In this paper, we revisit the vanilla full fine-tuning for VLMs and show that full fine-tuning is more efficient than prompt tuning under data-limited scenarios. To mitigate the overfitting and catastrophic forgetting issues encountered when fine-tuning the entire VLMs for specific tasks under limited supervision, we propose a framework named CLIP-CITE via designing a discriminative visual-text task, further aligning the visual-text semantics in a supervision manner, and integrating knowledge distillation techniques to preserve the gained knowledge. Extensive experimental results under few-shot learning, base-to-new generalization, domain generalization, and cross-domain generalization settings, demonstrate that our method effectively enhances the performance on specific tasks under limited supervision while preserving the versatility of the VLMs on other datasets.

## 1 INTRODUCTION

Recently, the pre-trained Vision-Language Models (VLMs) such as CLIP Radford et al. (2021) and ALIGN Jia et al. (2021) have demonstrated impressive generalization capabilities across various downstream tasks Zhou et al. (2022b); Gu et al. (2021); Rao et al. (2022); Rasheed et al. (2023). Though versatile, the performance of the VLMs on specific domains shows considerable potential for improvement, especially under data-limited scenarios Zhou et al. (2022b); Khattak et al. (2023a).

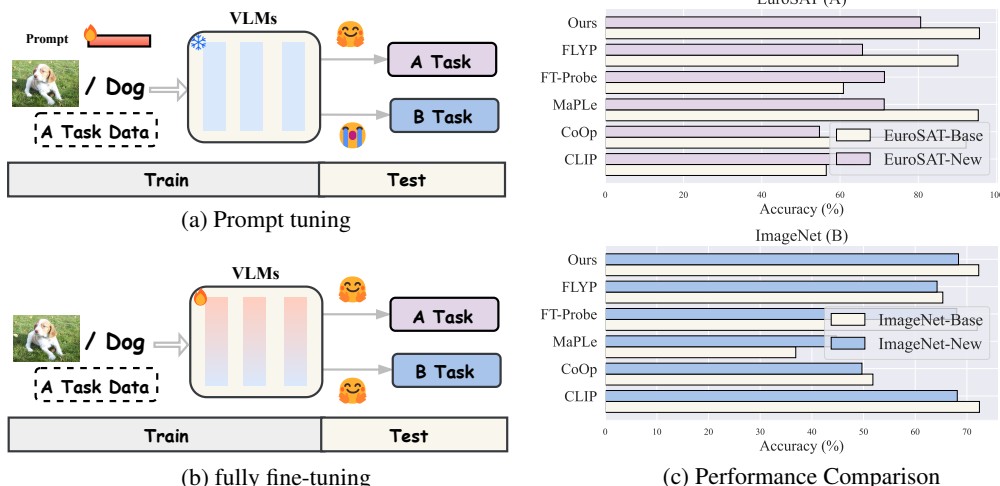

Figure 1: (a) Prompt tuning methods Zhou et al. (2022b); Khattak et al. (2023a). (b) Our full fine-tuning methods Radford et al. (2021); Goyal et al. (2023). (c) Comparison results (%) under the cross-domain generalization setting in the limited-data regime.

Prompt tuning Zhou et al. (2022b); Khattak et al. (2023a); Jia et al. (2022) has been proposed to refine pre-trained models using extra parameters while keeping the parameters of VLMs fixed. These methods gain popularity due to efficient parameter utilization and the ability to quickly adapt VLMs to domain-specific information in data-limited scenarios. While the prompt-based tuning strategies enable VLMs to effectively capture domain-specific information with limited supervision, there is a risk that these strategies may compromise the versatility of VLMs where the prompts trained on domain-specific data may struggle to generalize to different domains. A piece of evidence is provided in Fig. 1(c), which shows a transfer experiment under the cross-domain generalization setting. We employ a few-shot setting to train the model using the EuroSAT base training set, followed by evaluating its performance on both EuroSAT and ImageNet datasets. While the prompt-based methods, i.e., CoOp Zhou et al. (2022b) and MaPLe Khattak et al. (2023a) significantly improve the EuroSAT dataset results, they are at the cost of sacrificing their generalizability on the other datasets. In particular, their performances on the ImageNet dataset severely lag behind those of the original CLIP model.

Recent advances in natural language processing and text-to-image generation have demonstrated that fully fine-tuning can capture domain-specific information, even with limited data Liu et al. (2021); Ruiz et al. (2023). When customizing VLMs to specific domains, fine-tuning the entire models would distribute task-specific knowledge across all parameters (as illustrated in Fig. 1. (b)). To explore the effect of fully fine-tuning in adapting VLMs, we compare two existing methods: FT-Probe Radford et al. (2021), which fine-tunes the model while adding a linear probe, and FLYP Goyal et al. (2023), which fine-tunes using the contrastive loss from a pre-trained model. Our findings indicate that FT-Probe underperforms on the EuroSAT dataset, whereas FLYP achieves balanced performance across various datasets. From the results, we posit that the key to effective fine-tuning lies in determining the appropriate fine-tuning parameters and optimization objectives to transfer knowledge to specific domains while preserving the model's generalization capabilities. Therefore, in this paper, we revisit the fully fine-tuning paradigm for VLMs and propose that, with suitable optimization objectives and techniques, *fully fine-tuning the inherent parameters of VLMs can outperform existing parameter-efficient methods in data-limited scenarios*.

We propose a fine-tuning method called **CLIP-CITE** that enhances the CLIP's professionalism on specific domains while preserving its generalization by primarily enhan**C**ing the capability of the **I**mage-**T**ext alignm**E**nt task. CLIP-CITE fine-tunes all the parameters of both the text encoder and the image encoder, optimizing them to better capture domain-specific features. Additionally, CLIP-CITE incorporates three key aspects to optimize models. Firstly, to quickly equip the domain-specific information for CLIP, our CLIP-CITE connects the alignment score with the classification probability in a way that prioritizes higher alignment scores for image-text pairs belonging to the same class. Secondly, our approach fine-tunes the entire model using an image-text alignment task, aligning with the original training objective of the pre-trained CLIP model. This differs from the classification task utilized in Radford et al. (2021), ensuring a consistent training objective throughout the adaptation process. Note that training an image-text alignment task usually requires a large batch Radford et al. (2021); Goyal et al. (2023) in implementation, posing a significant challenge when working with limited data regimes. To overcome this issue, we propose utilizing a class-level image-text alignment task as an alternative to the original instance-level alignment task. Finally, to alleviate the catastrophic forgetting issue, we introduce a vision-language similarity distillation strategy. This strategy regularizes the model by transferring the image-text alignment relationship learned by the pre-trained CLIP model, further ensuring a minimal change in parameters. As shown in the last row of Fig. 1 (c), our CLIP-CITE enhances EuroSAT dataset performance while simultaneously upholding generalization capability on the ImageNet dataset.

In summary, our highlights are as follows:

- We revisit the fully fine-tuning in VLMs and propose a simple but efficient fine-tuning method to enhance the VLMs' professionalism while maintaining their versatility under limited data supervision. CLIP-CITE comprehensively fine-tunes CLIP to enable it to promptly incorporate task-specific information through enhanced image-text alignment and safeguard the learned knowledge.

- To verify the effectiveness of our proposed CLIP-CITE method, we conduct an analysis of the parameter changes across various layers of the fine-tuned model, employing diverse optimization functions and fine-tuning parameters.

- We evaluate CLIP-CITE through experiments in different settings, including base-to-new generalization, domain generalization, and cross-domain scenarios. The experimental results demonstrate that CLIP-CITE not only sets new benchmarks in these tasks on specific datasets but also preserves the original versatility of CLIP on other datasets.

## 2 RELATED WORK

**Vision-Language Models.** In recent years, significant advancements have been made in large-scale pre-trained vision-language models Radford et al. (2021); Jia et al. (2021); Wang et al. (2021); Alayrac et al. (2022); Wang et al. (2022); Huang et al. (2023). Representatively, CLIP Radford et al. (2021) and ALIGN Jia et al. (2021) jointly associate the images and their corresponding text descriptions by optimizing a contrastive objective. Training on the millions of image-text pairs, CLIP aligns the image and language space, showing the powerful generalization on downstream tasks. Based on CLIP, many works seek to transfer the model to special tasks, e.g., zero-shot recognition Lu et al. (2024), few-shot image recognition Zhou et al. (2022b;a); Khattak et al. (2023a), segmentation Rao et al. (2022), and action recognition Rasheed et al. (2023); Liu et al. (2024). In this paper, we leverage multi-modal alignment and the generalization ability of CLIP. We explore the potential of fully fine-tuning CLIP in limited-data scenarios.

**Few-Shot Transfer Learning Based on CLIP.** Prompt tuning Zhou et al. (2022b;a); Jia et al. (2022); Khattak et al. (2023a); Zhang et al. (2024); Wang et al. (2024) and fine-tuning Shu et al. (2023); Wortsman et al. (2022); Goyal et al. (2023); Kumar et al. (2022) are two main methods to transfer the CLIP to the downstream tasks. Prompt tuning is widely used in language models Houlsby et al. (2019); Liu et al. (2023), which raises attention in vision and multi-modality areas Zhou et al. (2022b); Jia et al. (2022); Kirillov et al. (2023). CoOp Zhou et al. (2022b) enhances downstream few-shot image recognition by learning soft textual prompts. Building on this, VPT Jia et al. (2022) and MaPLe Khattak et al. (2023a) explore visual and multi-modal prompts to further improve performance. PromptSRC Khattak et al. (2023b) and CoPrompt Roy & Etemad (2024) introduce regularization techniques for learnable prompts, promoting better generalization in novel scenarios. MetaPrompt Park et al. (2024) leverages meta-learning to optimize multi-modal prompts, adapting effectively to new tasks. Although these prompt tuning methods show efficient and excellent performance, they may fail to overfit the task-specific distribution. As the alternative, fine-tuning methods directly optimize the model under task-specific situations. WiSE-FT Wortsman et al. (2022), LP-FT Kumar et al. (2022) achieves the robustness of fine-tuning via a weight-ensemble manner. CLIPood Shu et al. (2023) further finetunes the model via the text semantic similarity and model ensemble under an out-of-distribution situation. A similar work related to our method is FLYP Goyal et al. (2023), which fine-tunes the CLIP model via the pre-trained contrastive objective to obtain the multi-modal alignment ability. In comparison, our method distinguishes the supervised vision-language pairs and incorporates the task-specific into the fine-tuning process. Leveraging this improved image-text alignment task, our method aims to perform more robustly under limited supervision.

## 3 METHOD

In this work, we fine-tune the CLIP model Radford et al. (2021) for the scenarios with limited data available. The architecture of CLIP includes two key components: a visual encoder denoted as $\theta_I$ and a text encoder denoted as $\theta_T$. By aligning language and visual modalities on 400 million text-image web data, CLIP is endowed with zero-shot and open-vocabulary capabilities.

To perform zero-shot classification, CLIP utilizes handcrafted text prompts with class labels. These prompts consist of a predefined set of class labels denoted as $y \in \{y_1, y_2, ..., y_C\}$, where $C$ represents the total number of classes. Each prompt typically takes the form of "a photo of a [category]", where "[category]" corresponds to the class label name. Then, the label prediction $\hat{y}$ of image $x$ corresponding to the class $c$ is obtained by calculating the cosine similarity scores between the image embedding $\mathbf{I}$ and the text embedding $\mathbf{T}$, which is formulated as:

$$p(\hat{y}|x) = \frac{\exp\left(s\left(\mathbf{I}, \mathbf{T}_c\right)/\tau\right)}{\sum_{i=1}^{C}\exp\left(s\left(\mathbf{I}, \mathbf{T}_i\right)/\tau\right)}, \tag{1}$$

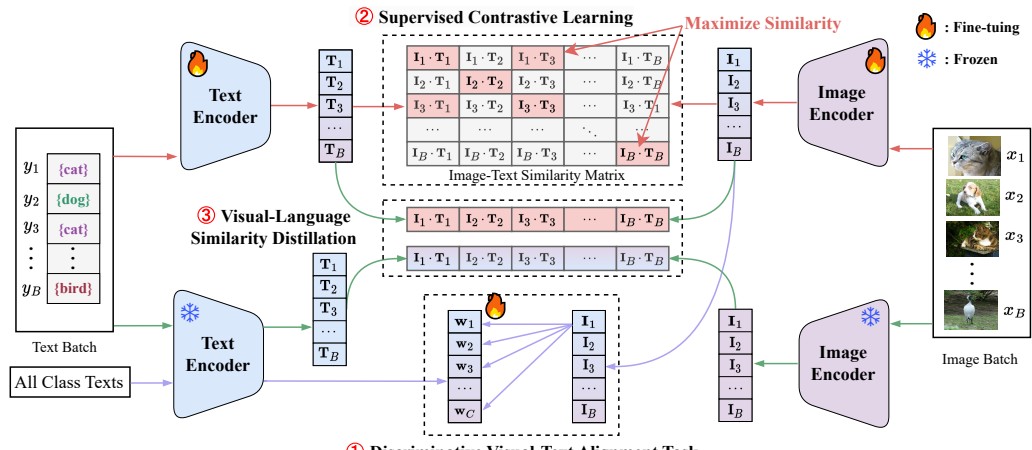

Figure 2: The framework of our CLIP-CITE method. CLIP-CITE fine-tunes the whole CLIP model with a ① discriminative visual-text alignment task and a ② supervised contrastive loss to enhance the image-text alignment in downstream tasks. Moreover, a ③ vision-language similarity distillation loss incorporates the generalization knowledge of the pre-trained CLIP model into the fine-tuned model.

where $s(\cdot)$ is the similarity metric, $\tau$ denotes the temperature parameter. By calculating the softmax probabilities using the similarity scores, CLIP assigns a class label to the image, even if it has not been explicitly trained on that specific class.

Although CLIP has demonstrated impressive zero-shot performance, its integration into specific downstream tasks still requires further refinements through subtle adjustments. Extensive prompt-based methods Zhou et al. (2022b;a); Khattak et al. (2023a) have been proposed to enhance CLIP's performance in specific contexts. In this study, we investigate the underestimated fine-tuning strategy and propose to improve the fine-tuning method from the perspectives of task designing, multi-modal alignment, and knowledge preservation. As illustrated in Fig. 2, our framework comprises three components, i.e., discriminative visual-text alignment task, supervised contrastive learning, and vision-language similarity distillation.

## 3.1 DISCRIMINATIVE VISUAL-TEXT ALIGNMENT TASK

Naive fine-tuning methods for downstream classification tasks typically involve adding a randomly initialized linear classifier on top of the pre-trained visual encoder Radford et al. (2021); Kumar et al. (2022). The whole model is then fine-tuned using the available domain-specific data for the classification task at hand. However, this training strategy often leads to overfitting on the limited available training data, resulting in poor generalization performance on unseen data.

To address this limitation, we propose to fine-tune the model with a discriminative visual-text alignment task that combines visual-semantic alignment and image classification. Specifically, we connect the similarity scores between the visual and the text embeddings with the probability that the visual image belongs to the class associated with the text embedding, which is formulated:

$$p(\hat{y}|x) = \frac{\exp\left(s\left(\theta_I\left(x\right), \theta_T\left(t_i\right)\right)\right)}{\sum_{c=1}^{C} \exp\left(s\left(\theta_I\left(x\right), \theta_T\left(t_c\right)\right)\right)}, \tag{2}$$

where $s(\cdot)$ is the consine similarity, $\theta_I$ and $\theta_T$ denotes the visual encoder and text encoder, respectively, $t_i$ is the text description of class $i$, which is obtained in the form of "a photo of a [category]", where "[category]" corresponds to one of the class labels.

Note that Eq. (2) is equivalent to initializing the parameters of the visual classifier $W = \{w_i\}_{i=0}^{C}, \ w_i = \theta_T\left(t_i\right)$ with the embeddings of the text descriptions of all the available classes and is consistent with the prediction of the test data. To this end, the objective function of the discriminative visual-text alignment task is:

$$\mathcal{L}_{DVA} = -\sum_{x \in \mathcal{B}} \log p(\hat{y}|x), \tag{3}$$

where $\mathcal{B}$ denotes a training batch during the fine-tuning process. To quickly adapt the model to the target classification task, we take $W$ and $\theta_I$ as the learnable parameter to fine-tune. Through fine-tuning this task, the model acquires the ability to collaboratively associate visual and textual representations, thereby enhancing its capacity to utilize semantic information effectively for the discriminative task.

## 3.2 SUPERVISED CONTRASTIVE LEARNING

To preserve and enhance the representation capability of the pre-trained CLIP, we argue that aligning image and text remains essential, as it corresponds to the task employed in the training of the original CLIP models. However, it is worth noting that aligning images and texts can often require a large batch size, which may not be suitable in situations where data availability is limited.

To mitigate this limitation, we customize an image-text alignment strategy to fine-tune the whole CLIP models (including both $\theta_I$ and $\theta_T$) under limited data regimes. Specifically, we adopt a supervised contrastive loss to align images and texts. Given a pair of data $(x, t)$, where $t$ is derived from the category of $x$ in the form of "a photo of a [category]", the supervised contrastive loss is defined as:

$$\mathcal{L}_{SCL} = - \sum_{x_i \in \mathcal{B}} \log \frac{\exp\left(s\left(\theta_I\left(x_i\right), \theta_T\left(t_i\right)\right)\right)}{\sum_{t_j \in \mathcal{B}} \mathbb{I}_{t_j \neq x_i} \cdot \exp\left(s\left(\theta_I\left(x_i\right), \theta_T\left(t_j\right)\right)\right)} \tag{4}$$
$$- \sum_{t_i \in \mathcal{B}} \log \frac{\exp\left(s\left(\theta_T\left(t_i\right)\right), \theta_I\left(x_i\right)\right)}{\sum_{x_j \in \mathcal{B}} \mathbb{I}_{x_j \neq t_i} \cdot \exp\left(s\left(\theta_T\left(t_i\right)\right), \theta_I\left(x_j\right)\right)},$$

where $s(,)$ denotes cosine similarity, $\mathcal{B}$ denotes a training batch, and $\mathbb{I}$ is the indicator function, defined as 1 if the image and text belong to the same class, and 0 otherwise. Notably, the loss function $\mathcal{L}_{SCL}$ can be viewed as a special case of the FLYP framework Goyal et al. (2023) in situations where no same-class instances are present within the batch, leveraging unsupervised contrastive loss to optimize the image-text alignment.

The designed supervised contrastive loss encourages the model to learn representations that bring similar images and their associated text embeddings closer together while pushing apart images and their non-matching text embeddings. By enforcing this alignment, the model can better capture the semantic relationship between images and their associated text while preserving and enhancing the representation capability of the pre-trained CLIP in specific domains.

## 3.3 VISION-LANGUAGE SIMILARITY DISTILLATION

While fine-tuning can improve performance on downstream tasks, it would suffer from potential challenges such as catastrophic forgetting and decreased generalization capabilities on the other datasets. To remedy this issue, we introduce a novel vision-language similarity distillation loss to distill the modal consistency from the pre-trained CLIP to the fine-tuned model. Specifically, the vision-language similarity distillation loss is defined as:

$$\mathcal{L}_{VLD} = \sum_{x \in \mathcal{B}} \mathcal{D}_{KL}\left(p\left(\hat{y}|x\right), \hat{p}\left(\hat{y}|x\right)\right), \tag{5}$$

where $p(\hat{y}|x)$ from Eq. (1) is computed using the fine-tuned models $\theta_I$ and $\theta_T$ to obtain batch cosine image-text similarity scores. Similarly, $\hat{p}(\hat{y}|x)$, also from Eq. (1), uses the original CLIP models. $\mathcal{D}_{KL}$ denotes the Kullback-Leibler divergence. The similarity scores are normalized using a softmax function to form a probability distribution.

By minimizing the Kullback-Leibler divergence between the distributions of image-text similarity calculated from the original CLIP encoders and those from the fine-tuned encoders, CLIP-CITE encourages the fine-tuned model to acquire comparable modal alignments and image-text relationship within batch as the pre-trained CLIP models. This strategy upholds modal consistency and facilitates the transfer of knowledge from the pre-trained model to the fine-tuned model.

## 3.4 FINAL OBJECTIVE FUNCTION

To fine-tune the whole CLIP models, we combine Eq. (3), Eq. (4), and Eq. (5), obtaining the final objective loss:

$$\mathcal{L} = \mathcal{L}_{DVA} + \lambda \cdot \mathcal{L}_{SCL} + \eta \cdot \mathcal{L}_{VLD}, \qquad (6)$$

where $\lambda$ and $\eta$ are the two hyperparameters to balance the items. After the fine-tuning process, we obtain the updated visual encoder $\theta_I$ and text encoder $\theta_T$.

During inference, we use a weighted ensemble proposed by Wortsman et al. (2022) to combine the fine-tuned model and the pre-trained model:

$$\hat{\theta}_I = \alpha \cdot \theta_I + (1 - \alpha) \cdot \theta_I^{zs}, \quad \hat{\theta}_T = \alpha \cdot \theta_T + (1 - \alpha) \cdot \theta_T^{zs}, \qquad (7)$$

where $\alpha$ is a hyperparameter. Different from Wortsman et al. (2022) that only considers ensemble in the visual modality, the text encoder in our method is optimized during the fine-tuning process, so the text modality is further considered in this work.

## 4 EXPERIMENTS

### 4.1 EXPERIMENT SETTINGS

**Benchmarks.** Following previous works Zhou et al. (2022a), we access the efficacy of our method within the few-shot learning paradigm via several popular benchmarks, *i.e.* Base-to-New Generation, Domain generalization, and Cross-Domain Generalization.

**Dataset Settings.** For Base-to-New Generalization, and Cross-Domain Generalization settings, we use 11 image classification datasets, i.e., ImageNet Deng et al. (2009) and Caltech-101 Fei-Fei et al. (2004) for generic object classification; OxfordPets Parkhi et al. (2012), StanfordCars Krause et al. (2013), Flowers Nilsback & Zisserman (2008), Food101 Bossard et al. (2014), and FGVCAircraft Maji et al. (2013) for fine-grained visual categorization, EuroSAT Helber et al. (2019) for satellite image classification, UCF101 Soomro et al. (2012) for action recognition, DTD Cimpoi et al. (2014) for texture classification, and SUN397 Xiao et al. (2010) for scene recognition. For the DG, we treat the ImageNet as the source domain, and the ImageNetV2 Recht et al. (2019), ImageNet-Sketch Wang et al. (2019), ImageNet-A Hendrycks et al. (2021b) and ImageNet-R Hendrycks et al. (2021a) as the target domains for evaluation.

**Implement Details.** We utilize the pre-trained ViT-B/16 of CLIP Radford et al. (2021); Dosovitskiy et al. (2020) as backbone. The initial learning rate is set to 5e-6 with the cosine annealing strategy and the batch size is set to 32 for most datasets. The hyperparameter $\lambda$ is set to 0.7, $\eta$ is set to 0.1, and $\alpha$ is set to 0.5. And the epoch is set to 20 for a trade-off. More datasets and experimental details are provided in Appendix A.1.

### 4.2 PERFORMANCE COMPARISON

**Results of Base-to-New Generalization.** Tab. 1 showcases the performance of our CLIP-CITE in comparison to CLIP Radford et al. (2021), CoOp Zhou et al. (2022b), CoCoOp Zhou et al. (2022a), ProDA Lu et al. (2022), MaPLe Khattak et al. (2023a), PromptSRC Khattak et al. (2023b), CoPrompt Roy & Etemad (2024), and CLIPFit Li et al. (2024). The accuracy metrics are reported for both the base classes (**B**), new classes (**N**), and their harmonic mean (**HM**). Our CLIP-CITE achieves the best performance on both B and N metrics averaged over 11 datasets, resulting in a 0.58% improvement in the HM metric over the second-best method. Compared to the original CLIP model without downstream fine-tuning, CLIP-CITE improves base class accuracy by 16.14% and novel class accuracy by 3.86%, demonstrating enhanced domain-specific performance and generalization in open-vocabulary settings. This indicates that fine-tuning with base data significantly enhances CLIP's capabilities, as also observed in other methods. However, while CoOp and CoCoOp improve base accuracy, they compromise generalization to novel classes. CLIP-CITE achieves the best HM metric on 7 out of 11 datasets compared to competitors, excelling particularly on fine-grained datasets like EuroSAT, Cars, and FGVCAircraf. This suggests that fine-tuning enables the model to capture more specialized information. We conclude that our method effectively mitigates overfitting and catastrophic forgetting, improving performance on both base and novel classes simultaneously.

Table 1: Comparison with state-of-the-art methods on base-to-new generalization. All methods use ViT-B/16 as the vision encoder. The results are reported for various datasets. HM stands for the harmonic mean.

| Method | Average | | | ImageNet | | | Caltech101 | | | OxfordPets | | |
|---|---|---|---|---|---|---|---|---|---|---|---|---|
| | Base | New | HM | Base | New | HM | Base | New | HM | Base | New | HM |
| CLIP | 69.34 | 74.22 | 71.70 | 72.43 | 68.14 | 70.22 | 96.84 | 94.00 | 95.40 | 91.17 | 97.26 | 94.12 |
| CoOp | 82.69 | 63.22 | 71.66 | 76.47 | 67.88 | 71.92 | 98.00 | 89.81 | 93.73 | 93.67 | 95.29 | 94.47 |
| CoCoOp | 80.47 | 71.69 | 75.83 | 75.98 | 70.43 | 73.10 | 97.96 | 93.81 | 95.84 | 95.20 | 97.69 | 96.43 |
| ProDA | 81.56 | 72.30 | 76.65 | 75.40 | 70.23 | 72.72 | 98.27 | 93.23 | 95.68 | 95.43 | 97.83 | 96.62 |
| MaPLe | 82.28 | 75.14 | 78.55 | 76.66 | 70.54 | 73.47 | 97.74 | 94.36 | 96.02 | 95.43 | 97.76 | 96.58 |
| PromptSRC | 84.26 | 76.10 | 79.97 | 77.60 | 70.73 | 74.01 | 98.10 | 94.03 | 96.02 | 95.33 | 97.30 | 96.30 |
| CoPrompt | 84.00 | 77.23 | 80.48 | 77.67 | 71.27 | 74.33 | 98.27 | 94.90 | 96.55 | 95.67 | 98.10 | 96.87 |
| CLIPFit | 83.72 | 74.84 | 79.03 | 76.20 | 70.17 | 73.06 | 98.30 | 93.70 | 95.94 | 95.23 | 97.13 | 96.17 |
| CLIP-CITE | 85.48 | 77.08 | 81.06 | 78.44 | 71.07 | 74.58 | 98.82 | 94.28 | 96.50 | 96.01 | 97.95 | 96.97 |

| Method | StanfordCars | | | Flowers102 | | | Food101 | | | FGVCAircraft | | |
|---|---|---|---|---|---|---|---|---|---|---|---|---|
| | Base | New | HM | Base | New | HM | Base | New | HM | Base | New | HM |
| CLIP | 63.37 | 74.89 | 68.65 | 72.08 | 77.80 | 74.83 | 90.10 | 91.22 | 90.66 | 27.19 | 36.29 | 31.09 |
| CoOp | 78.12 | 60.40 | 68.13 | 97.60 | 59.67 | 74.06 | 88.33 | 82.26 | 85.19 | 40.44 | 22.30 | 28.75 |
| CoCoOp | 70.49 | 73.59 | 72.01 | 94.87 | 71.75 | 81.71 | 90.70 | 91.29 | 90.99 | 33.41 | 23.71 | 27.74 |
| ProDA | 74.70 | 71.20 | 72.91 | 97.70 | 68.68 | 80.66 | 90.30 | 88.57 | 89.43 | 36.90 | 34.13 | 35.46 |
| MaPLe | 72.94 | 74.00 | 73.47 | 95.92 | 72.46 | 82.56 | 90.71 | 92.05 | 91.38 | 37.44 | 35.61 | 36.50 |
| PromptSRC | 78.27 | 74.97 | 76.58 | 98.07 | 76.50 | 85.95 | 90.67 | 91.53 | 91.10 | 42.73 | 37.87 | 40.15 |
| CoPrompt | 76.97 | 74.40 | 75.66 | 97.27 | 76.60 | 85.71 | 90.73 | 92.07 | 91.40 | 40.20 | 39.33 | 39.76 |
| CLIPFit | 78.80 | 73.87 | 76.26 | 96.83 | 73.53 | 83.59 | 90.60 | 91.33 | 90.96 | 42.47 | 33.47 | 37.43 |
| CLIP-CITE | 82.83 | 74.51 | 78.45 | 95.98 | 76.45 | 85.11 | 90.81 | 91.55 | 91.18 | 47.26 | 38.37 | 42.35 |

| Method | SUN397 | | | DTD | | | EuroSAT | | | UCF101 | | |
|---|---|---|---|---|---|---|---|---|---|---|---|---|
| | Base | New | HM | Base | New | HM | Base | New | HM | Base | New | HM |
| CLIP | 69.36 | 75.35 | 72.23 | 53.24 | 59.90 | 56.37 | 56.48 | 64.05 | 60.03 | 70.53 | 77.50 | 73.85 |
| CoOp | 80.60 | 65.89 | 72.51 | 79.44 | 41.18 | 54.24 | 92.19 | 54.74 | 68.69 | 84.69 | 56.05 | 67.46 |
| CoCoOp | 79.74 | 76.86 | 78.27 | 77.01 | 56.00 | 64.85 | 87.49 | 60.04 | 71.21 | 82.33 | 73.45 | 77.64 |
| ProDA | 78.67 | 76.93 | 77.79 | 80.67 | 56.48 | 66.44 | 83.90 | 66.00 | 73.88 | 85.23 | 71.97 | 78.04 |
| MaPLe | 80.82 | 78.70 | 79.75 | 80.36 | 59.18 | 68.16 | 94.07 | 73.23 | 82.35 | 83.00 | 78.66 | 80.77 |
| PromptSRC | 82.67 | 78.47 | 80.52 | 83.37 | 62.97 | 71.75 | 92.90 | 73.90 | 82.32 | 87.10 | 78.80 | 82.74 |
| CoPrompt | 82.63 | 80.03 | 81.31 | 83.13 | 64.73 | 72.79 | 94.60 | 78.57 | 85.84 | 86.90 | 79.57 | 83.07 |
| CLIPFit | 81.97 | 78.17 | 80.02 | 81.97 | 63.50 | 71.56 | 93.33 | 71.07 | 80.69 | 85.23 | 77.30 | 81.07 |
| CLIP-CITE | 82.30 | 79.40 | 80.82 | 84.26 | 64.54 | 73.09 | 95.61 | 80.59 | 87.46 | 87.56 | 79.01 | 83.07 |

**Results of Domain Generalization.** The DG performances of our method, along with six competitors, are presented in Tab. 2. In this evaluation, the model is trained on the few-shot ImageNet dataset and then tested on different datasets, namely ImageNetv2, ImageNet-Sketch, ImageNet-A, and ImageNet-R, which have the same class labels as ImageNet but belong to different domains. Our method demonstrates superior performance in terms of in-domain ImageNet accuracy, achieving an accuracy of 72.9%. Additionally, our method achieves a high average accuracy of 60.7% across the out-of-domain datasets, surpassing all existing methods except for ImageNet-A. These results indicate that our method is effective in handling domain shifts.

Table 2: Performance on domain generalization.

| Method | Source | Target | | | | |
|---|---|---|---|---|---|---|
| | ImNet | ImNetV2 | ImNetS | ImNetA | ImNetR | Ave. |
| CLIP | 66.73 | 60.83 | 46.15 | 47.77 | 73.96 | 57.17 |
| CoOp | 71.51 | 64.20 | 47.99 | 49.71 | 75.21 | 59.28 |
| Co-CoOp | 71.02 | 64.07 | 48.75 | 50.63 | 76.18 | 59.90 |
| MaPLe | 70.72 | 64.07 | 49.15 | **50.90** | 76.98 | 60.26 |
| PromptSRC | 71.27 | 64.35 | 49.55 | **50.90** | **77.80** | 60.65 |
| CoPrompt | 70.80 | 64.25 | 49.43 | 50.50 | 77.51 | 60.42 |
| CLIP-CITE | **72.90** | **65.80** | **49.60** | 50.00 | 77.50 | **60.70** |

**Results of Cross-Domain Generalization.** Unlike previous work Zhou et al. (2022a) that trains on ImageNet and evaluates on other datasets, we evaluate the model in a more challenge scenario where using training data from various datasets and evaluate it on the ImageNet test set. For ease of comparison with the results presented in Tab. 3, we report **HM** performance metrics on the ImageNet dataset. The results are shown in Tab. 3. CLIP-CITE maintains its performance on the ImageNet

Table 3: Cross-domain generalization evaluation (%). All models are trained on the base training set of 10 datasets and evaluated on the ImageNet dataset. Note that vanilla CLIP achieves 70.26% in terms of HM metric on ImageNet.

| Method | Caltech101 | OxfordPets | Cars | Flowers102 | Food101 | Aircrafts | SUN397 | DTD | EuroSAT | UCF101 |
|---|---|---|---|---|---|---|---|---|---|---|
| CoOP | 55.17 | 53.97 | 52.30 | 41.66 | 43.92 | 58.98 | 53.56 | 56.40 | 50.76 | 41.15 |
| CoCoOp | 68.83 | 63.98 | 57.70 | 53.25 | 64.46 | 56.28 | 68.42 | 63.92 | 59.58 | 53.00 |
| MaPLe | 71.01 | 55.37 | 67.39 | 53.00 | 69.82 | 64.97 | 68.91 | 64.78 | 42.95 | 63.29 |
| CLIP-CITE | **70.88** | **70.39** | **70.67** | **70.33** | **70.91** | **70.46** | **70.59** | **70.21** | **70.26** | **70.68** |

dataset regardless of the datasets used for training, indicating its robustness. In contrast, other competitors exhibit significant performance drops on ImageNet. For example, the **HM** performance on ImageNet falls from 70.22% to 42.95% when fine-tuning the model with MaPLe Khattak et al. (2023a) on the EuroSAT dataset. We attribute this to parameter-efficient competitors capturing domain- and class-specific information, rendering them less suitable for novel classes from different domains. In contrast, our fully fine-tuning method distributes the changes in domain and category equally across the parameters of the model, resulting in small changes in parameter magnitude, which enables it to effectively handle different domains and categories simultaneously. Furthermore, our distillation strategy also benefits in mitigating catastrophic forgetting.

### 4.3 FINE-TUNING ANALYSIS

**Effects of Fine-tuning Parts.** In this experiment, we conduct an ablation study to examine the effects of different fine-tuning parts. The average results of 11 datasets and the results on ImageNet dataset are shown in Fig. 3. **FrozenTC** indicates that the text embeddings are taken as the classifiers of the visual feature representations and are frozen during optimizing Eq. (3). **FrozenTE** indicates that the text encoder is frozen during optimizing Eq. (4). **ALL** indicates that all the parameters of the model are fine-tuning during training. From the results in Fig. 3, we observe that HM performance of **ALL** witnesses a considerable lift compared with those of **FrozenTC** and **FrozenTE**, which concludes that comprehensive fine-tuning enhances model capabilities more effectively than partial fine-tuning.

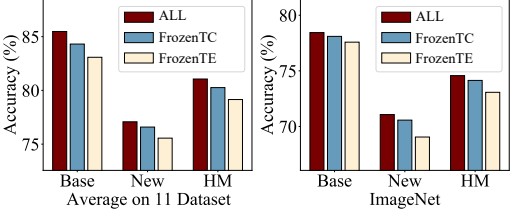

Figure 3: The effect of different fine-tuning parts of the model. Fine-tuning all parts achieves the best performance on ImageNet Datasets, and shows a similar trend across 11 datasets.

Figure 4: Visualization of parameter changes in different layers. (a) refers to the parameter changes of the text encoder. (b) indicates the parameter changes of the image encoder in different layers.

**Parameter Analysis.** To investigate the impact of fine-tuning on CLIP-CITE, we compute the squared differences of parameters at each layer and conducted experiments using the EuroSAT dataset, as shown in Fig. 4. The results demonstrate that as the layer depth increases, the parameter changes in both the image encoder and the text encoder gradually diminish. We attribute this phenomenon to the larger gradient propagation in earlier layers, which leads to more significant parameter updates. This trend was observed in both the image and text encoders. Furthermore, when we introduced our designed loss functions, $\mathcal{L}_{SCL}$ and $\mathcal{L}_{VLD}$, the parameter changes were somewhat mitigated, suggesting that optimizing the loss functions can impose regularization constraints on the model to a certain extent. This finding further corroborates the critical role of $\mathcal{L}_{SCL}$ and $\mathcal{L}_{VLD}$ losses in fine-tuning downstream tasks. We discuss the effect of fine-tuning layers later.

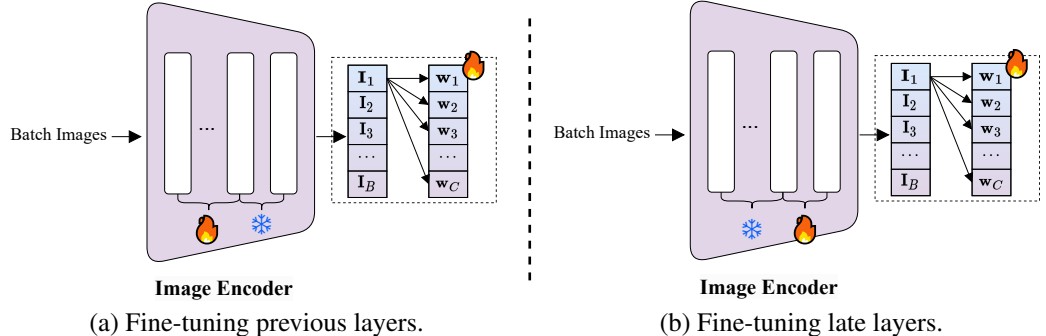

(a) Fine-tuning previous layers.                    (b) Fine-tuning late layers.

Figure 5: Illustration of the fine-tuned model within the distinct layers. (a) illustrates layers preceding the image encoder, while (b) delineates layers succeeding the image encoder.

**Effects of Fine-tuning Layers.** Fig. 5 shows the different fine-tuning manners of the image encoder, *e.g.* fine-tuning previous layers and fine-tuning late layers. And we conduct the experiments with $\mathcal{L}_{DVA}$ for ablation. Fig. 6. (a) shows results where we fine-tune previous layers and freeze late layers, while Fig. 6. (b) shows results where we freeze previous layers and fine-tune late layers. From the experimental results, we observe that when there are only a few frozen layers, the

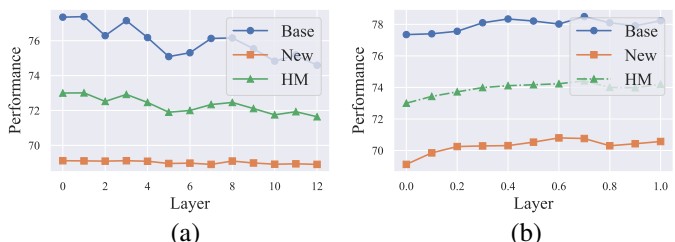

Figure 6: The effect of fine-tuning layers. (a) indicates we fine-tune the previous layers and freeze the $i_{th}$ late layers corresponding to Fig. 5.(a), while (b) indicates we freeze the previous $i_{th}$ layers and fine-tune the late layers corresponding to Fig. 5. (b).

performance is comparable to fully fine-tuning. However, as the number of frozen layers increases, the effectiveness diminishes, *i.e.* the last 3 frozen layers led to a decline in the results shown in Fig. 6. (a). Overall, fully fine-tuning is better than partial fine-tuning.

**Fine-tuning Training Efficiency.** Tab. 4 presents a comprehensive comparison of our CLIP-CITE and four parameter-efficient competitors. The results indicate that parameter efficiency does not necessarily translate to computational efficiency. Specifically, our model, despite fine-tuning more parameters and utilizing more GPU resources, demonstrates superior performance with significantly fewer training iterations and shorter overall training time compared to the parameter-efficient competitors. Although prompt-based methods offer parameter efficiency, they still require the backpropagation of the entire model, along with numerous training iterations, to achieve convergence.

More analyses are provided in the Appendix A.2.

Table 4: Comparison performances (%) and training efficiency of the existing prompt tuning methods and ours. All the models are trained on a single NVIDIA GeForce RTX 3090 GPU.

| Method | Iterations | ImageNet | | | Training Resources | |
|---|---|---|---|---|---|---|
| | | Base | New | HM | Training-time | GPU-usage |
| CLIP | N/A | 72.43 | 68.14 | 70.22 | $N/A$ | $N/A$ |
| CoOp | 12.5 K | 76.47 | 67.88 | 71.92 | $\approx 1\,h$ | $\approx 10\,G$ |
| CoCoOp | 80K | 75.98 | 70.43 | 73.10 | $> 7\,h$ | $\approx 10\,G$ |
| MaPLe | 10 K | 76.66 | 70.54 | 73.47 | $\approx 45\,min$ | $\approx 10\,G$ |
| CLIP-CITE | 1.2K | **78.44** | **71.07** | **74.58** | $\approx 20\,min$ | $\approx 19\,G$ |

## 4.4 ABLATION STUDIES

**Effects of Different Objectives.** Sec. 4.4 displays the ablation study of our CLIP-CITE with various training objectives on the Base-to-New task of ImageNet. The first row represents the results obtained with the basic CLIP model. When fine-tuning the model with only $\mathcal{L}_{DVA}$, it achieves a 2.78% improvement in **HM** compared to the naive CLIP. Additionally, the introduction of supervised contrastive learning objective $\mathcal{L}_{SCL}$ leads to further improvement in both **B** and **N** metrics. By combining both objectives ($\mathcal{L}_{DVA} + \mathcal{L}_{SCL}$), the performance of both **B** and **H** metrics continue to improve. Furthermore, incorporating the vision-language similarity distillation loss $\mathcal{L}_{VLD}$ into the objective results in the best performance of 74.58% **HM** accuracy. Notably, we also compare FLYP Goyal et al. (2023) with our proposed method. While FLYP demonstrates improved performance over the original CLIP, it falls short of achieving the objectives of our design. We attribute this to FLYP's exclusive focus on image-text alignment, which overlooks downstream tasks and hinders its effectiveness in data-limited scenarios. These experimental outcomes highlight the efficacy of each objective function introduced in this work.

Table 5: Ablation results (%) of our CLIP-CITE with various training objectives on the Base-to-New task of the ImageNet dataset. † denotes that we re-implement with the official code.

| Method | B | N | HM |
|---|---|---|---|
| CLIP | 72.43 | 68.14 | 70.22 |
| FLYP† | 76.21 | 68.13 | 71.94 |
| CLIP-CITE (+$\mathcal{L}_{DVA}$) | 77.35 | 69.12 | 73.00 |
| CLIP-CITE (+$\mathcal{L}_{SCL}$) | 78.10 | 70.67 | 74.20 |
| CLIP-CITE (+$\mathcal{L}_{DVA}, \mathcal{L}_{SCL}$) | **78.49** | 70.76 | 74.43 |
| CLIP-CITE (+$\mathcal{L}_{DVA}, \mathcal{L}_{VLD}$) | 77.31 | 70.20 | 73.58 |
| CLIP-CITE (+$\mathcal{L}_{DVA}, \mathcal{L}_{SCL}, \mathcal{L}_{VLD}$) | 78.44 | **71.07** | **74.58** |

**Effects of Weight Ensemble.** We investigate the effect of weight ensemble in Fig. 7. The results therein lead us to the conclusion that even without the weight ensemble inference (with $\alpha$ set to 1.0), our method still delivers noteworthy performance with results of 85.79% (B), 73.52% (N), and 79.19% (HM) on the Base-to-New task. Notably, it outperforms CLIPood, which integrates model weight inference ensemble, and MaPLe, by achieving a lift of 0.27% and 0.64% in the HM metric, respectively. Moreover, with the appropriate weight ensemble ratio (setting $\alpha$ to 0.5), we have noticed a notable improvement in both base and novel performance.

More results of ablation studies are provided in the Appendix A.3.

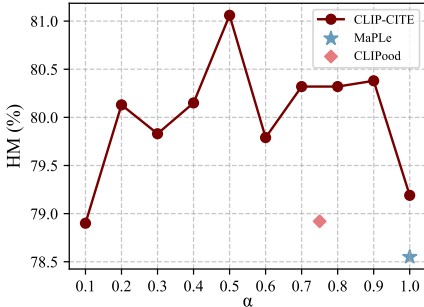

Figure 7: Comparison results with the different ensemble ratio $\alpha$.

## 5 CONCLUSION

In this paper, we have presented CLIP-CITE, a fully fine-tuning approach designed to adapt CLIP for downstream tasks in limited-data scenarios. By devising a discriminative visual-text alignment task, implementing supervised contrastive loss, and employing visual-language similarity distillation, CLIP-CITE effectively addresses the common issues of overfitting and catastrophic forgetting encountered by existing fine-tuning methods. Our experimental results demonstrate that a carefully crafted fine-tuning strategy can enable CLIP to acquire both domain-specific and class-specific knowledge, while maintaining its versatility across other domains and classes. Notably, despite involving the tuning of more parameters, our approach offers superior computational efficiency compared to parameter-efficient prompt-based competitors. We hope this work can promote in-depth research on the full fine-tuning paradigm of VLMs in data-limited scenarios and facilitate its future application to recent Multimodal Large Language Models (MLLMs).

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

## A APPENDIX

This section contains supplementary material that provides additional details for the main paper and further experimental analysis. The content of this section is as follows:

- Additional Experimental Details
- Additional Experimental Analysis
- Additional Ablation Study

### A.1 ADDITIONAL EXPERIMENTAL DETAILS

**Dataset Details.** In Tab. 6, we list the details of the datasets and the hand-crafted prompt we used in the experiments. The prompts are from the Radford et al. (2021) and we have not adopted more prompt templates to generate the optical text representations. In this work, we only focus on the effect of fully fine-tuned CLIP and the text representations would be automatically learned during the training.

**Training Details.** We maintain the temperature of the softmax function consistent with the pre-trained model, using $\tau = 0.01$, except for when $\mathcal{L}_{VLD}$ is adjusted to 0.1. All images are randomly resized and cropped to $224 \times 224$, only random resize and random crop data augments are applied. The optical hyper-parameter $\lambda$ is set to 0.7, $\eta$ is set to 0.1, and $\alpha$ is set to 0.5 for all experiments. We use the AdamW optimizer with the cosine learning rate strategy and the learning rate is set to 5e-6 and trained for 20 epochs. The batch size is set to 32 for most datasets, with specific batch sizes of 16 for EuroSAT and 64 for ImageNet. For each result of CLIP-CITE, we report the average result with three random seeds.

Table 6: Detailed statistics of the datasets.

| Dataset | Classes | Train | Val | Test | Hand-crafted Prompt |
|---|---|---|---|---|---|
| Caltech101 | 100 | 4,128 | 1,649 | 2,465 | a photo of a [CLS]. |
| OxfordPets | 37 | 2,944 | 736 | 3,669 | a photo of a [CLS], a type of pet. |
| StanfordCars | 196 | 6,509 | 1,635 | 8,041 | a photo of a [CLS]. |
| Flowers102 | 102 | 4,093 | 1,633 | 2,463 | a photo of a [CLS], a type of flower. |
| Food101 | 101 | 50,500 | 20,200 | 30,300 | a photo of [CLS], a type of food. |
| FGVCAircraft | 100 | 3,334 | 3,333 | 3,333 | a photo of a [CLS], a type of aircraft. |
| SUN397 | 397 | 15,880 | 3,970 | 19,850 | a photo of a [CLS]. |
| DTD | 47 | 2,820 | 1,128 | 1,692 | [CLS] texture. |
| EuroSAT | 10 | 13,500 | 5,400 | 8,100 | a centered satellite photo of [CLS]. |
| UCF101 | 101 | 7,639 | 1,898 | 3,783 | a photo of a person doing [CLS]. |
| ImageNet | 1,000 | 1.28M | N/A | 50,000 | a photo of a [CLS] |
| ImageNetV2 | 1,000 | N/A | N/A | 10,000 | a photo of a [CLS] |
| ImageNet-Sketch | 1,000 | N/A | N/A | 50,889 | a photo of a [CLS] |
| ImageNet-A | 200 | N/A | N/A | 7,500 | a photo of a [CLS] |
| ImageNet-R | 200 | N/A | N/A | 30,000 | a photo of a [CLS] |

### A.2 ADDITIONAL EXPERIMENTAL ANALYSIS

**Overfitting Analysis.** We demonstrate the training process of **FT-Probe** and our **CLIP-CITE** illustrated in Fig. 1 on EuroSAT dataset. The results of loss and accuracy of the training dataset are shown in Fig. 8. We observe that, for the FT-Probe model, there is a decline in the training loss, accompanied by a continual increase in accuracy on the training set. However, the final accuracy on the test set is only 60.86%, which suggests the occurrence of overfitting. In contrast, in the case of our CLIP-CITE model, there is also a reduction in the loss function and a consistent rise in training set accuracy, culminating in a test set accuracy of 95.61%. This indicates that our approach does not exhibit overfitting, demonstrating effectiveness. Moreover, it highlights that overcoming overfitting is a crucial issue when fully fine-tuning models.

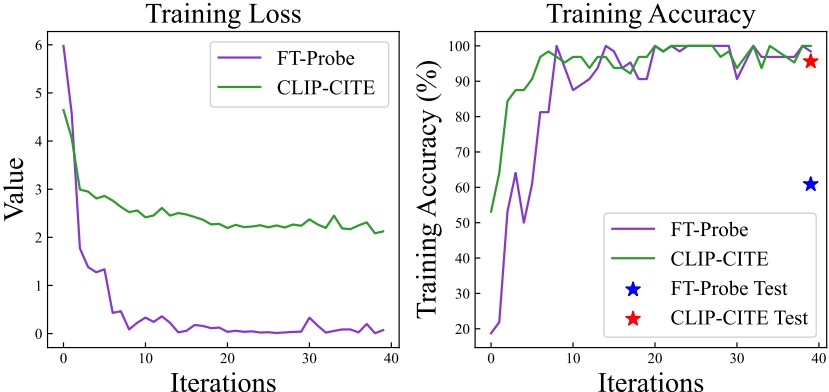

Figure 8: Training loss and accuracy of FT-Probe and CLIP-CITE on EuroSAT dataset.

## A.3 Additional Ablation Study

**Prompt Tuning with Proposed Loss.** To evaluate the effectiveness of full-fine-tuning, we also explore the prompt tuning methods with our proposed loss. The results, detailed in Tab. 7, indicate that prompt tuning methods experience a modest improvement with the implementation of our proposed loss functions *i.e.* $L_{SCL}$ and $L_{VLD}$. Notably, our CLIP-CITE still maintains a performance edge. Besides, with the simple fine-tuning (FT-Probe), the tuned model seems to be overfitting, as shown in Fig. 1. Therefore, we propose that both full fine-tuning and well-designed loss functions are crucial in adapting VLMs to the downstream few-shot tasks.

Table 7: Ablation results (%) of our CLIP-CITE and prompt tuning, and fine-tuning methods with various training objectives on the Base-to-New of the ImageNet dataset.

| Method | $\mathcal{L}_{SCL}$ | $\mathcal{L}_{VLD}$ | B | N | HM |
|---|---|---|---|---|---|
| CLIP | | | 72.43 | 68.14 | 70.22 |
| FLYP † | | | 76.21 | 68.13 | 71.94 |
| CoOp | | | 76.47 | 67.88 | 71.92 |
| CoOp | ✓ | | 76.51 | 67.93 | 71.97 |
| CoOp | ✓ | ✓ | 78.23 | 70.89 | 72.11 |
| MaPLe | | | 76.66 | 70.54 | 73.47 |
| MaPLe | ✓ | | 76.70 | 70.67 | 73.56 |
| MaPLe | ✓ | ✓ | 76.71 | 70.89 | 73.69 |
| CLIP-CITE | ✓ | ✓ | 78.44 | 71.07 | 74.58 |

**The Effect of the Hyper-Parameter $\lambda$ and $\eta$.** In Fig. 9, we ablate the different values on $\lambda$ and $\eta$ in Eq. (6). From the results, we observe that the performances in terms of HM are better when applying the $\mathcal{L}_{SCL}$, e.g., $\lambda$ is greater than 0. It indicates that supervised vision-language alignment is necessary when fine-tuning. Besides, the vision-language similarity distillation can regularize the model well when $\eta$ is less than 0.1. In the experiments, the optical $\lambda$ and $\eta$ are set to 0.7 and 0.1, respectively.

**Results of Few-Shot Image Recognition.** Fig. 10 presents the average results of four competitors and our CLIP-CITE on the 11 datasets under 1, 2, 4, 8, and 16 shots. From the results, we observe that our CLIP-CITE performs very competitively, especially under 1, 2, and 4 shots. When compared with the second-best competitor MaPLe Khattak et al. (2023a) on the average results, our CLIP-CITE demonstrates performance improvements by 3.42%, 3.00%, 2.48%, 1.73%, and 1.52% in scenarios with 1, 2, 4, 8, and 16 shots, respectively. These gains underscore CLIP-CITE's effectiveness in generalizing to downstream tasks when provided with limited labeled examples. More comparisons of each dataset are provided in the supplementary materials.

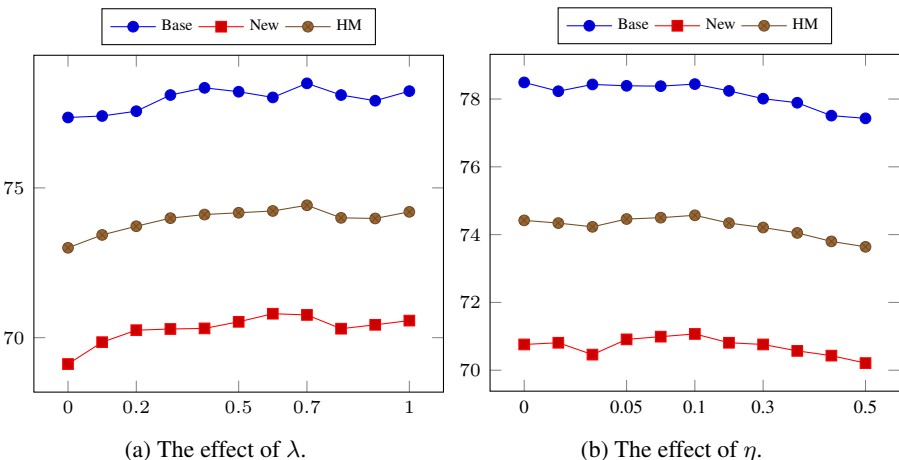

(a) The effect of $\lambda$.        (b) The effect of $\eta$.

Figure 9: The impacts of the hyper-parameter $\lambda$ and $\eta$ on the base-to-new generalization performances. We report the Base (%), New (%), and HM (%) accuracy on the ImageNet dataset.

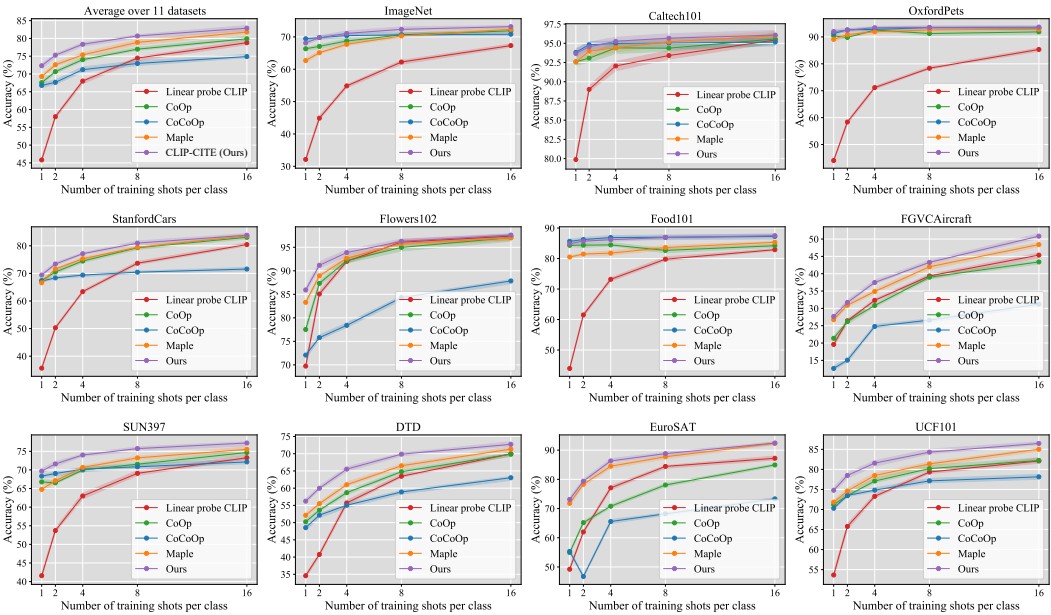

Figure 10: Comparison results of few-shot learning benchmark on the 11 datasets. All of the methods are trained on the ViT-B/16 backbone and implemented with the same experimental settings.

**The Effect of Weights Ensemble Ratio $\alpha$.** Tab. 9 shows the results of different datasets with the different ensemble ratios $\alpha$. Without weights ensemble, CLIP-CITE achieves 85.79%, 73.52%, and 79.19% in Base, New, and HM accuracy, respectively. With the fine-tuning weights ensemble, the performance increases from 71.70% to 78.90% in HM accuracy when $\alpha$ is 0.1. When $\alpha$ increases, the Base accuracy increases, and the New accuracy fluctuates slightly. The optimal value $\alpha$ appears to be 0.5. This indicates that our fine-tuning process maintains a subtle change of model parameters, facilitating smooth compatibility with the zero-shot pre-trained CLIP model and resulting in an overall enhancement of effectiveness.

**More Experimental Results of Cross-Domain Generalization Setting.** Tab. 8 and Tab. 3 shows the experimental results of Cross-Domain setting. From the results of Tab. 8, all methods trained on the ImageNet can consistently obtain the generalization performance on the other 10 datasets. From the results of Tab. 3, the prompt tuning methods trained on other datasets are difficult to transfer to ImageNet and impact the overall generalization, while our fine-tuning methods can maintain or even

| Methods | CLIP | | CoOp | | CoCoOp | | MaPLe | | CLIP-CITE | |
|---|---|---|---|---|---|---|---|---|---|---|
| | Base | New | Base | New | Base | New | Base | New | Base | New |
| Caltech101 | 96.84 | 94.00 | 94.15 | 93.92 | 96.58 | 95.16 | 96.30 | 94.98 | 96.71 | 93.82 |
| OxfordPets | 91.17 | 97.26 | 90.34 | 97.69 | 90.8 | 97.97 | 90.57 | 97.73 | 89.56 | 96.78 |
| StanfordCars | 63.37 | 74.89 | 61.99 | 73.37 | 63.62 | 74.48 | 62.44 | 73.98 | 60.74 | 72.41 |
| Flowers102 | 72.08 | 77.80 | 66.86 | 75.23 | 72.30 | 77.64 | 73.25 | 76.86 | 71.48 | 76.9 |
| Food101 | 90.10 | 91.22 | 88.62 | 90.68 | 89.39 | 91.0 | 89.29 | 90.86 | 88.47 | 90.53 |
| FGVCAircraft | 27.19 | 36.29 | 21.21 | 26.36 | 27.65 | 32.37 | 28.69 | 31.21 | 26.33 | 34.33 |
| SUN397 | 69.36 | 75.35 | 68.36 | 72.78 | 72.08 | 75.96 | 71.46 | 76.1 | 71.78 | 76.16 |
| DTD | 53.24 | 59.90 | 49.00 | 51.73 | 55.32 | 57.01 | 51.04 | 54.51 | 50.39 | 57.53 |
| EuroSAT | 56.48 | 64.05 | 50.2 | 69.22 | 52.1 | 68.84 | 47.52 | 59.83 | 49.50 | 65.51 |
| UCF101 | 70.53 | 77.50 | 68.89 | 71.88 | 70.89 | 75.77 | 69.22 | 74.97 | 70.99 | 76.55 |

Table 8: Cross-Domain evaluation. All the models are trained on the base training set of the ImageNet dataset and evaluated on the 10 datasets .

| $\alpha$ ratio | | 0 | 0.1 | 0.2 | 0.3 | 0.4 | **0.5** | 0.6 | 0.7 | 0.8 | 0.9 | 1.0 |
|---|---|---|---|---|---|---|---|---|---|---|---|---|
| | Base | 69.34 | 83.53 | 84.66 | 84.44 | 85.09 | 85.48 | 85.64 | 85.69 | 85.64 | 85.58 | 85.79 |
| Average on | New | 74.22 | 74.75 | 76.07 | 75.71 | 75.74 | 77.08 | 74.68 | 75.58 | 75.62 | 75.77 | 73.52 |
| | HM | 71.70 | 78.90 | 80.13 | 79.83 | 80.15 | 81.06 | 79.79 | 80.32 | 80.32 | 80.38 | 79.19 |
| | Base | 72.43 | 77.45 | 77.63 | 78.20 | 78.23 | 78.44 | 78.44 | 78.48 | 78.46 | 78.49 | 78.50 |
| ImageNet | New | 68.14 | 70.35 | 70.79 | 70.71 | 70.65 | 71.07 | 70.32 | 70.59 | 70.36 | 70.29 | 70.23 |
| | HM | 70.22 | 73.73 | 74.05 | 74.27 | 74.25 | 74.58 | 74.16 | 74.33 | 74.19 | 74.17 | 74.14 |
| | Base | 96.84 | 97.20 | 97.65 | 97.78 | 98.77 | 98.82 | 98.82 | 98.83 | 98.83 | 98.83 | 98.85 |
| Caltech101 | New | 94.00 | 93.40 | 94.14 | 93.44 | 93.65 | 94.28 | 93.53 | 93.90 | 94.00 | 93.47 | 93.20 |
| | HM | 95.40 | 95.26 | 95.86 | 95.56 | 96.14 | 96.50 | 96.10 | 96.30 | 96.36 | 96.08 | 95.94 |
| | Base | 91.17 | 95.23 | 95.84 | 95.66 | 95.82 | 96.01 | 96.18 | 96.42 | 96.60 | 96.93 | 97.01 |
| OxfordPets | New | 97.26 | 96.12 | 96.47 | 96.69 | 96.71 | 97.95 | 96.66 | 96.72 | 96.90 | 97.28 | 95.23 |
| | HM | 94.12 | 95.67 | 96.15 | 96.17 | 96.27 | 96.97 | 96.42 | 96.57 | 96.75 | 97.11 | 96.11 |

Table 9: Comparison with the different ensemble ratio $\alpha$ on base-to-new generalization.

enhance the performance of ImageNet. These demonstrate that ImageNet encompasses a broader array of patterns and categories, and both prompt tuning methods and our approach effectively sustain performance across various datasets. When transferring from other datasets to ImageNet, CLIP-CITE can uphold ImageNet's performance. It shows that our fine-tuning method has better generalization capacity.

| Dataset | Method | 1-shot | 2-shot | 4-shot | 8-shot | 16-shot |
|---|---|---|---|---|---|---|
| Average | Linear probe CLIP | 45.83 | 57.98 | 68.01 | 74.47 | 78.79 |
| | CoOp | 67.56 | 70.65 | 74.02 | 76.98 | 79.89 |
| | CoCoOp | 66.79 | 67.65 | 71.21 | 72.96 | 74.90 |
| | MaPLe | 69.27 | 72.58 | 75.37 | 78.89 | 81.79 |
| | CLIP-CITE | 72.69 | 75.58 | 77.85 | 80.62 | 83.31 |
| ImageNet | Linear probe CLIP | 32.13 | 44.88 | 54.85 | 62.23 | 67.31 |
| | CoOp | 66.33 | 67.07 | 68.73 | 70.63 | 71.87 |
| | CoCoOp | 69.43 | 69.78 | 70.39 | 70.63 | 70.83 |
| | MaPLe | 62.67 | 65.10 | 67.70 | 70.30 | 72.33 |
| | CLIP-CITE | 68.20 | 68.90 | 70.30 | 71.20 | 72.90 |
| Caltech101 | Linear probe CLIP | 79.88 | 89.01 | 92.05 | 93.41 | 95.43 |
| | CoOp | 92.60 | 93.07 | 94.4 | 94.37 | 95.57 |
| | CoCoOp | 93.83 | 94.82 | 94.98 | 95.04 | 95.16 |
| | MaPLe | 92.57 | 93.97 | 94.43 | 95.2 | 96.00 |
| | CLIP-CITE | 94.16 | 94.81 | 95.53 | 96.39 | 96.50 |
| OxfordPets | Linear probe CLIP | 44.06 | 58.37 | 71.17 | 78.36 | 85.34 |
| | CoOp | 90.37 | 89.8 | 92.57 | 91.27 | 91.87 |
| | CoCoOp | 91.27 | 92.64 | 92.81 | 93.45 | 93.34 |
| | MaPLe | 89.10 | 90.87 | 91.9 | 92.57 | 92.83 |
| | CLIP-CITE | 91.47 | 93.02 | 93.54 | 93.87 | 94.70 |
| StanfordCars | Linear probe CLIP | 35.66 | 50.28 | 63.38 | 73.67 | 80.44 |
| | CoOp | 67.43 | 70.5 | 74.47 | 79.3 | 83.07 |
| | CoCoOp | 67.22 | 68.37 | 69.39 | 70.44 | 71.57 |
| | MaPLe | 66.60 | 71.60 | 75.30 | 79.47 | 83.57 |
| | CLIP-CITE | 70.63 | 74.22 | 76.53 | 79.94 | 83.70 |
| Food101 | Linear probe CLIP | 43.96 | 61.51 | 73.19 | 79.79 | 82.90 |
| | CoOp | 84.33 | 84.40 | 84.47 | 82.67 | 84.20 |
| | CoCoOp | 85.65 | 86.22 | 86.88 | 86.97 | 87.25 |
| | MaPLe | 80.50 | 81.47 | 81.77 | 83.60 | 85.33 |
| | CLIP-CITE | 85.16 | 85.95 | 86.05 | 86.68 | 87.00 |
| Flowers102 | Linear probe CLIP | 69.74 | 85.07 | 92.02 | 96.10 | 97.37 |
| | CoOp | 77.53 | 87.33 | 92.17 | 94.97 | 97.07 |
| | CoCoOp | 72.08 | 75.79 | 78.40 | 84.30 | 87.84 |
| | MaPLe | 83.30 | 88.93 | 92.67 | 95.80 | 97.00 |
| | CLIP-CITE | 84.25 | 86.76 | 92.08 | 95.86 | 97.6 |
| FGVCAircraft | Linear probe CLIP | 19.61 | 26.41 | 32.33 | 39.35 | 45.36 |
| | CoOp | 21.37 | 26.20 | 30.83 | 39.00 | 43.40 |
| | CoCoOp | 12.68 | 15.06 | 24.79 | 26.61 | 31.21 |
| | MaPLe | 26.73 | 30.90 | 34.87 | 42.00 | 48.40 |
| | CLIP-CITE | 29.34 | 32.40 | 36.60 | 46.00 | 57.00 |
| SUN397 | Linear probe CLIP | 41.58 | 53.70 | 63.00 | 69.08 | 73.28 |
| | CoOp | 66.77 | 66.53 | 69.97 | 71.53 | 74.67 |
| | CoCoOp | 68.33 | 69.03 | 70.21 | 70.84 | 72.15 |
| | MaPLe | 64.77 | 67.10 | 70.67 | 73.23 | 75.53 |
| | CLIP-CITE | 69.54 | 70.99 | 72.36 | 74.45 | 76.30 |
| DTD | Linear probe CLIP | 34.59 | 40.76 | 55.71 | 63.46 | 69.96 |
| | CoOp | 50.23 | 53.60 | 58.70 | 64.77 | 69.87 |
| | CoCoOp | 48.54 | 52.17 | 55.04 | 58.89 | 63.04 |
| | MaPLe | 52.13 | 55.50 | 61.00 | 66.50 | 71.33 |
| | CLIP-CITE | 54.20 | 60.70 | 64.54 | 67.67 | 72.50 |
| EuroSAT | Linear probe CLIP | 49.23 | 61.98 | 77.09 | 84.43 | 87.21 |
| | CoOp | 54.93 | 65.17 | 70.80 | 78.07 | 84.93 |
| | CoCoOp | 55.33 | 46.74 | 65.56 | 68.21 | 73.32 |
| | MaPLe | 71.80 | 78.30 | 84.50 | 87.73 | 92.33 |
| | CLIP-CITE | 76.20 | 85.20 | 88.77 | 91.17 | 92.60 |
| UCF101 | Linear probe CLIP | 53.66 | 65.78 | 73.28 | 79.34 | 82.11 |
| | CoOp | 71.23 | 73.43 | 77.10 | 80.20 | 82.23 |
| | CoCoOp | 70.30 | 73.51 | 74.82 | 77.14 | 78.14 |
| | MaPLe | 71.83 | 74.60 | 78.47 | 81.37 | 85.03 |
| | CLIP-CITE | 76.40 | 78.38 | 80.07 | 83.56 | 85.70 |

Table 10: Per-dataset performance comparison of our method with various methods in the few-shot setting.