# OpenReview forum: "Fully Fine-Tuning Beats Parameter Efficient Fine-Tuning for CLIP in Data-Limited Scenarios"
_ICLR.cc/2025/Conference — ICLR 2025 Conference Withdrawn Submission_

### Official Review · Reviewer_utnR · 2024-10-27

**Soundness:** 2
**Presentation:** 2
**Contribution:** 2
**Rating:** 3
**Confidence:** 4

**Summary:**

This paper introduces a fully fine-tuning framework, CLIP-CITE, designed to adapt CLIP for data-limited scenarios in downstream tasks. The method combines three components: Discriminative Visual-Text Alignment Task, Supervised Contrastive Learning, and Vision-Language Similarity Distillation. These modules aim to enhance alignment and prevent overfitting, thereby improving performance across domain and cross-domain settings.

**Strengths:**

The paper is well-organized,  making it easy to follow the methodology and experimental results.

**Weaknesses:**

1. The motivation is unclear. The paper looks like the authors tried some scattered techniques without a strong motivation. For example, what is the connection between task designing, multi-modal alignment, and knowledge preservation?

2. Lack of Novelty: The proposed components—Discriminative Visual-Text Alignment Task, Supervised Contrastive Learning, and Vision-Language Similarity Distillation—are primarily applications of existing techniques within vision-language models (VLMs). These components do not introduce significant technological innovations and lack strong motivation for why they are specifically needed for the proposed task. For example, the Discriminative Visual-Text Alignment Task builds upon standard techniques already employed in VLM training, i.e., the vanilla contrastive loss, without providing a unique improvement or adaptation that would set it apart from previous work.

3. Lack of Justification for Effectiveness: The paper does not adequately demonstrate why the proposed methods are effective for addressing the specific challenges it mentions.  For example, why can the DISCRIMINATIVE VISUAL-TEXT ALIGNMENT TASK alleviate the problem of poor generalization performance? What is the principle?  What are the advantages of SUPERVISED CONTRASTIVE LEARNING?

4. Insufficient Theoretical Support: The methodology relies heavily on empirical results, with limited theoretical foundation to support the effectiveness of the proposed alignment and similarity distillation strategies. A theoretical analysis, such as discussing the mathematical or functional impact of the new losses on fine-tuning stability or domain adaptation, would be beneficial to justify the approach. This would help establish why these strategies are more effective for maintaining performance across domains compared to alternative fine-tuning or parameter-efficient techniques.

**Questions:**

See the Weakness

---

### Official Review · Reviewer_CYZo · 2024-10-28

**Soundness:** 3
**Presentation:** 3
**Contribution:** 2
**Rating:** 5
**Confidence:** 4

**Summary:**

This paper presents CLIP-CITE via , a fine-tune paradigm of VLMs via designing a discriminative visual-text task and aligning the visual-text semantics in a supervision manner, which additionally integrates knowledge distillation techniques to preserve the gained knowledge. Extensive experiment results demonstrate the effectiveness of the proposed method.

**Strengths:**

CLIP-CITE enhances the CLIP’s professionalism on specific domains while preserving its generalization by primarily enhancing the capability of the Image-Text alignment task.

Extensive experimental results under few-shot learning, base-to-new generalization, domain generalization, and crossdomain generalization settings are conducted to demonstrate the effectiveness of the method.

**Weaknesses:**

1. Supervised contrastive learning [1] is not new. What is the difference between the supervised contrastive learning in CLIP-CITE and [1]?
2. Several state-of-the-art prompt tuning approaches are missing in comparisons (e.g., TCP[2]).

*Reference*

[1] Khosla, Prannay, et al. "Supervised contrastive learning." Advances in neural information processing systems 33 (2020): 18661-18673.

[2] Yao, Hantao, Rui Zhang, and Changsheng Xu. "TCP: Textual-based Class-aware Prompt tuning for Visual-Language Model." Proceedings of the IEEE/CVF Conference on Computer Vision and Pattern Recognition. 2024.

**Questions:**

In table 4, how about the performance of different approaches under the same interations? It would be better to additionally compare the performance under the same iterations for fair comparison.

---

### Official Review · Reviewer_KLWy · 2024-11-02

**Soundness:** 2
**Presentation:** 2
**Contribution:** 2
**Rating:** 3
**Confidence:** 4

**Summary:**

This work proposes CLIP-CITE, a fine-tuning method to enhance CLIP on specific domains and expect to preserve the generalization ability. This is achieved by enhance the capability of the image-text alignment task. The proposed CLIP-CITE fine-tunes the parameters not only of the text-encoder but also the image encoder, aiming to better capture the domain-specific features.

**Strengths:**

This work proposes a simple method that fine-tunes the VLMs to enhance its professionalism and maintains the versatility in cases where the data is limited.
The experiments verifies the effectiveness of the proposed method in different settings, includings base-to-new, domain generalization and cross-domain scenarios.

**Weaknesses:**

1. The proposed method is not well motivated. Please discuss the motivations of the proposed method in details.
2. In the cases where the data is limited. Are the sufficient to fine-tune both the text-encoder and the image-encoder? How many data should we have at the minimum?
3. Please provide more implementation details and present more experimental results.

**Questions:**

1. The proposed method is not well motivated. Please discuss the motivations of the proposed method in details.
2. In the cases where the data is limited. Are the sufficient to fine-tune both the text-encoder and the image-encoder? How many data should we have at the minimum?
3. Please provide more implementation details and present more experimental results.

---

### Official Review · Reviewer_2iNH · 2024-11-03

**Soundness:** 3
**Presentation:** 4
**Contribution:** 2
**Rating:** 3
**Confidence:** 5

**Summary:**

This paper proposes a new fine-tuning method, CLIP-CITE, which fully fine-tunes Vision-Language Models (VLMs), specifically CLIP, to improve performance in data-limited scenarios. The authors argue that, contrary to the recent trend of parameter-efficient tuning, full fine-tuning yields better results under limited supervision. They introduce a framework with three key components—Discriminative Visual-Text Alignment, Supervised Contrastive Learning, and Vision-Language Similarity Distillation—to mitigate overfitting and catastrophic forgetting while fine-tuning the entire model. Experimental results demonstrate that CLIP-CITE outperforms state-of-the-art parameter-efficient methods across various tasks, including few-shot learning, base-to-new generalization, and cross-domain generalization.

**Strengths:**

- The proposed method covers multiple aspects of the fine-tuning process, including contrastive learning, task-specific alignment, and distillation, showing a good grasp of fine-tuning challenges in data-limited scenarios.
- The paper includes multiple benchmarks and tasks (e.g., base-to-new generalization, domain, and cross-domain generalization) to validate the efficacy of CLIP-CITE.
- The technical components and experimental details are explained clearly, and the paper is easy to follow.

**Weaknesses:**

The paper has a few key limitations.
- To begin with, I find the problem statement to be a bit weak. Specifically, this part: "There is a risk that these strategies may compromise the versatility of VLMs where the prompts trained on domain-specific data may struggle to generalize to different domains." I don't see solid evidence to support this claim. Even the toy example in Figure 1c does not quite tell the story. The proposed solution performs just as well as the pre-trained CLIP.
- Also, the paper mostly fails to acknowledge that some papers have already explored the issue that fine-tuning comes at the cost of lower generalization.
- Even the proposed method has two major problems:
    - (a) Limited novelty. The components are minor modifications/adaptations to the particular task. Concepts like supervised contrastive learning, distillation or visual-text alignment have no new contribution.
    - (b) None of these concepts are specific to full fine-tuning. Why can't we just use them on the best PEFT method out there? Will it not improve the performance? If not, why?

**Questions:**

See the weakness section.

---

### Note · Authors · 2024-11-15

I have read and agree with the venue's withdrawal policy on behalf of myself and my co-authors.